



**Indirect contributions of global fires to surface ozone through**
**ozone-vegetation feedback**
**Yadong Lei[1,2], Xu Yue[3*], Hong Liao[3], Lin Zhang[4], Yang Yang[3], Hao Zhou[1,2],**
**Chenguang Tian[1,2], Cheng Gong[2,5], Yimian Ma[1,2], Lan Gao[1,2], Yang Cao[1,2]**
[1]Climate Change Research Center, Institute of Atmospheric Physics, Chinese
Academy of Sciences, Beijing, 100029, China
[2]University of Chinese Academy of Sciences, Beijing, 100029, China
[3]Jiangsu Key Laboratory of Atmospheric Environment Monitoring and Pollution
Control, Collaborative Innovation Center of Atmospheric Environment and
Equipment Technology, School of Environmental Science and Engineering, Nanjing
University of Information Science & Technology (NUIST), Nanjing, 210044, China
[4]Laboratory for Climate and Ocean–Atmosphere Studies, Department of Atmospheric
and Oceanic Sciences, School of Physics, Peking University, Beijing, 100871, China
[5]State Key Laboratory of Atmospheric Boundary Layer Physics and Atmospheric
Chemistry (LAPC), Institute of Atmospheric Physics, Chinese Academy of Sciences,
Beijing, 100029, China
*Correspondence to*: Xu Yue (yuexu@nuist.edu.cn)



**Abstract:** Fire is an important source of surface ozone ($O_3$), which causes damage to
vegetation and reduces stomatal conductance. Such processes can feed back to inhibit
dry deposition and indirectly enhance surface $O_3$. Here, we apply a fully coupled
chemistry-vegetation model to estimate the indirect contributions of global fires to
surface $O_3$ through $O_3$-vegetation feedback during 2005-2012. Fire emissions directly
increase the global mean annual $O_3$ by 1.2 ppbv (5.0%) with a maximum of 5.9 ppbv
(24.4%) averaged over central Africa by emitting substantial number of precursors.
Considering $O_3$-vegetation feedback, fires additionally increase surface $O_3$ by 0.5
ppbv averaged over the Amazon in October, 0.3 ppbv averaged over southern Asia in
April, and 0.2 ppbv averaged over central Africa in April. During extreme
$O_3$-vegetation interactions, such feedback can rise to >0.6 ppbv in these fire-prone
areas. Moreover, large ratios of indirect-to-direct fire $O_3$ are found in eastern China
(3.7%) and the eastern U.S. (2.0%), where the high ambient $O_3$ causes strong
$O_3$-vegetation interactions. With likelihood of increasing fire risks in a warming
climate, fires may promote surface $O_3$ through both direct emissions and indirect
chemistry-vegetation feedbacks. Such indirect enhancement will cause additional
threats to public health and ecosystem productivity.

**Keywords:** fires, surface ozone, dry deposition, ozone-vegetation feedback





## 1 Introduction


Fire plays an important role in disturbing the terrestrial carbon budget
(Bond-Lamberty et al., 2007; Amiro et al., 2009; Turetsky et al., 2011; Yue and Unger,
2018). Global fires directly emit 2-3 Pg (1 Pg = $10^{15}$ g) carbon into the atmosphere
every year (van der Werf et al., 2010). Moreover, fires contribute to the production of
tropospheric ozone ($O_3$) by emitting substantial number of precursors (Cheng et al.,
1998; Kita et al., 2000; Oltmans et al., 2010; Jaffe et al., 2013; Lu et al., 2016).
Globally, fires account for 3-5% of the total tropospheric $O_3$ (Bey et al., 2001; Ziemke
et al., 2009; Jaffe and Wigder, 2012). Regionally, the influence of fires on $O_3$
production is dependent on mixing with urban emissions (Jaffe et al., 2004; Singh et
al., 2010). In some areas, fires can enhance surface $O_3$ by 10-30 ppbv through
emissions of $NO_x$ and VOCs (McKeen et al., 2002; Pfister et al., 2008; Yue and Unger,
2018). Model simulations project that future wildfire activity will likely increase due
to global warming, suggesting an increased risk of surface $O_3$ from wildfires (Amiro
et al., 2009; Balshi et al., 2009; Wang et al., 2016; Yue et al., 2017).

Tropospheric $O_3$ is a toxic air pollutant with detrimental effects on vegetation (Yue
and Unger, 2014). Plant stomatal uptake of $O_3$ decreases both chlorophyll and
Rubisco contents and increases the deformity rate of chloroplasts (Booker et al., 2007;
Akhtar et al., 2010; Inada et al., 2012), which further reduces the leaf area index (LAI)
and gross primary productivity (GPP) of ecosystems (Karnosky et al., 2007;
Ainsworth et al., 2012). Modeling studies estimated that fire-induced $O_3$ reduces



global GPP by 0.7% with regional maximum reductions of >4.0% over central Africa
(Yue and Unger, 2018). In turn, vegetation influences both the sources and sinks of $O_3$
through biogeochemical and biogeophysical feedbacks (Curci et al., 2009; Heald and
Geddes, 2016; Fitzky et al., 2019). Emissions from biomass burning generate a large
amount of $O_3$ precursors (Jaffe and Wigder, 2012; Lu et al., 2016). Moreover,
vegetation acts as an important sink for tropospheric $O_3$ through stomatal uptake
(Wesely and Hicks, 2000; Val Martin et al., 2014). Globally, stomatal uptake
contributes to 40-60% of the canopy total $O_3$ deposition (Fowler et al., 2009).

Interactions between air pollution and terrestrial ecosystems remain challenging due
to limited process-based knowledge and the separate development of chemistry and
vegetation models (He et al., 2020). At present, the feedbacks from $O_3$-damaging
vegetation on $O_3$ have only been examined by three papers. By implementing
steady-state $O_3$-induced LAI changes into a chemical transport model, Zhou et al.
(2018) quantified the influences of $O_3$-vegetation feedback and found that $O_3$-induced
damage to LAI can enhance $O_3$ by up to 3 ppbv in the tropics, eastern North America,
and southern China. Moreover, plant stomatal conductance may decrease to prevent
excessive $O_3$ from entering plants (Manninen et al., 2003; Wittig et al., 2009).
Consequently, surface $O_3$ may increase due to reduced dry deposition (Val Martin et
al., 2014; Lin et al., 2019). Sadiq et al. (2017) implemented a parameterization of $O_3$
vegetation damage into a climate model and quantified online $O_3$-vegetation coupling.
Simulation results showed that surface $O_3$ can be enhanced by up to 4-6 ppbv over



Europe, North America, and China mainly because of reduced dry deposition velocity
following $O_3$ damage. Similarly, Gong et al. (2020) used a fully coupled
chemistry-carbon-climate global model and found that $O_3$-induced inhibition of
stomatal conductance can increase surface $O_3$ by 1.4-2.1 ppbv in eastern China and
1.0-1.3 ppbv in western Europe. All studies revealed strong positive $O_3$-vegetation
feedback to surface $O_3$, although the magnitudes are different due to discrepancies in
$O_3$ damaging schemes, as well as differences in the climate models.

Many studies have quantified the direct contributions of fires to tropospheric $O_3$
(Martin et al., 2006; Pfister et al., 2006; Ziemke et al., 2009; Yokelson et al., 2011;
Jaffe and Wigder, 2012; Larsen et al., 2018; Yue and Unger, 2018). However, the
feedback of fire-induced $O_3$ vegetation damage to surface $O_3$ remain unquantified.
Here, we apply a fully coupled chemistry-vegetation model (GEOS-Chem-YIBs,
hereafter referred to as GC-YIBs) to examine the indirect contributions of fires to
surface $O_3$. Fire-induced $O_3$ affects plant photosynthesis and stomatal conductance. In
turn, predicted changes in LAI and canopy stomatal conductance influence both the
sources and sinks of tropospheric $O_3$. Such $O_3$-vegetation interactions result in
additional enhancement in surface $O_3$ caused by fire emissions (Fig. 1). Section 2
describes the GC-YIBs model and sensitivity experiments conducted in this study.
Section 3 quantifies the feedbacks of fire-induced $O_3$ vegetation damage on surface
$O_3$ concentrations. The last section summarizes the findings and discusses the
uncertainties.




## 2 Materials and Methods

### 2.1 The GC-YIBs model

GC-YIBs is a coupled chemistry-vegetation model developed by implementing the

Yale Interactive terrestrial Biosphere (YIBs) model into GEOS-Chem version 12.0.0

(Lei et al., 2020). GEOS-Chem is a widely used global 3-D chemical transport model

(CTM) for simulating atmospheric composition and air quality (Yue et al., 2015; Yan

et al., 2018; David et al., 2019; Lu et al., 2019). This model uses a detailed

$HO_x$-$NO_x$-VOC-$O_3$-halogen-aerosol tropospheric chemistry to simulate tropospheric

$O_3$ fluxes (Barret et al., 2016; Gong and Liao, 2019), while a simplified linearized

Linoz chemistry mechanism is applied to simulate stratospheric $O_3$ (McLinden et al.,

2000). Aerosols simulated in GEOS-Chem include secondary inorganic aerosols,

secondary organic aerosols, primary organic aerosols, black carbon, dust, and sea salt

(Dang and Liao, 2019; Li et al., 2019). The gas-aerosol partitioning of the

sulfate–nitrate–ammonium system is computed by the ISORROPIA v2.0

thermodynamic equilibrium model (Fountoukis and Nenes, 2007). The atmospheric

emissions from different sources, regions, and species on a user-defined grid are

calculated through the online Harvard NASA Emissions Component (HEMCO)

module (Keller et al., 2014). HEMCO is highly customizable in that it can

automatically combinate, overlay, and update emission inventories and scale factors

specified by the users. In general, the GEOS-Chem model overestimates summer

surface $O_3$ concentrations in the eastern U.S. and China (Zhang et al., 2011; Travis et



al., 2016; Schiferl and Heald, 2018).

YIBs is a vegetation model designed to dynamically simulate the changes in LAI and
tree height based on carbon assimilation, respiration, and allocation processes (Yue
and Unger, 2015). The model computes carbon uptake for 9 vegetation types,
including evergreen needleleaf forest, deciduous broadleaf forest, evergreen broadleaf
forest, shrubland, tundra, $C_3$/$C_4$ grasses, and $C_3$/$C_4$ crops. The YIBs model applies a
well-established Michaelis–Menten enzyme kinetics scheme to compute the leaf
photosynthesis for $C_3$ and $C_4$ plants (Farquhar et al., 1980; Von Caemmerer and
Farquhar, 1981). The leaf stomatal conductance was calculated based on the model of
Ball and Berry (Baldocchi et al., 1987). The Spitters (1986) canopy radiative transfer
scheme is used to separate light use processes for sunlit and shaded leaves. The LAI
and carbon allocation schemes are from the TRIFFID model (Clark et al., 2011).
Previous studies have shown that the YIBs model has good performance in simulating
the spatial pattern and temporal variability of GPP and LAI based on site observations
and satellite products (Yue and Unger, 2015, 2018).

The GC-YIBs model links atmospheric chemistry and vegetation in a two-way
coupling. As a result, changes in chemical components or vegetation will
simultaneously feed back to influence the other systems. In this study, the GC-YIBs
model is driven with the meteorological fields from the Modern-Era Retrospective
analysis for Research and Applications, version 2 (MERRA2) with a horizontal





resolution of 4° latitude by 5° longitude, as well as 47 vertical layers from the surface
to 0.01 hPa. Within GC-YIBs, the online-simulated surface $O_3$ in GEOS-Chem affects
photosynthesis and canopy stomatal conductance; in turn, the online-simulated
vegetation parameters, such as LAI and stomatal conductance, in YIBs, affect both the
sources and sinks of $O_3$ by altering precursor emissions and dry deposition at the
1-hour integration time step. An earlier study evaluated the GC-YIBs model and
showed good performance in simulating surface $O_3$, GPP, LAI, and $O_3$ dry deposition
(Lei et al., 2020).

**2.2 Scheme of $O_3$ vegetation damage**
The GC-YIBs model calculates the impacts of $O_3$ exposure on photosynthesis based
on a semi-mechanistic scheme (Sitch et al., 2007):
$$A^{'} = \alpha \cdot A \qquad (1)$$
where  $A^{'}$  and  $A$  represent the $O_3$-damaging and original leaf photosynthesis,
respectively. The $O_3$ damage factor is represented by $\alpha$; $O_3$ can cause damage to
photosynthesis only if $\alpha < 1$. The factor $\alpha$ is calculated as a function of excessive $O_3$
flux and damaging sensitivity coefficient ($\beta$):
$$\alpha = -\beta \cdot max\big(F_{O_3} - T_{O_3}, 0\big) \qquad (2)$$
The coefficient $\beta$ can have two values for each vegetation type (Table S1), indicating
low to high $O_3$ damaging sensitivities (Sitch et al., 2007). $T_{O_3}$  represents the $O_3$ flux
threshold, reflecting the $O_3$ tolerance of different vegetation types. $F_{O_3}$  represents the
stomatal $O_3$ flux and is calculated based on ambient  $[O_3]$, aerodynamic resistance



($r_a$), boundary layer resistance ($r_b$) and stomatal resistance ($r_s$):
$$F_{O_3} = \frac{[O_3]}{r_a + r_b + k \cdot r_s{}'}$$
(3)

Here $k$ represents the ratio of leaf resistance for $O_3$ to leaf resistance for water vapor.
Parameters $r_a$ and $r_b$ are calculated by the GEOS-Chem model. $O_3$-damaging leaf
photosynthesis ($A'$) is then integrated over all canopy layers to generate $O_3$-damaging
GPP:

$$GPP' = \int_0^{LAI} A' \, dL$$

The $O_3$-damaging stomatal resistance ($r_s{}'$) is calculated based on the model of Ball
and Berry (Baldocchi et al., 1987):
$$\frac{1}{r_s{}'} = g_s{}' = m \frac{A'_{net} \cdot RH}{c_s} + b$$
(4)

where $m$ and $b$ represent the slope and intercept of empirical fitting to the Ball-Berry
stomatal conductance equation, respectively. $A'_{net}$ represents $O_3$-damaging net leaf
photosynthesis, $RH$ represents the relative humidity and $c_s$ is the ambient $CO_2$
concentration. Previous studies have shown that this scheme within the framework of
YIBs can reasonably capture the response of GPP and stomatal conductance to
surface [$O_3$] based on hundreds of global observations (Yue et al., 2016; Yue and
Unger, 2018).

**2.3 Fire emissions**
Fire Inventory from NCAR (FINN) version 1.5 is used by GC-YIBs to simulate
fire-induced perturbations in $O_3$. FINN provides daily global emissions of many
chemical species from open biomass burning at a resolution of 1 km$^2$ (Wiedinmyer et



al., 2011). The inventory estimates fire locations and biomass burned using satellite
observations of active fires and land cover, together with emission factors and fuel
loadings. For each land type, emission factors for different gaseous and particulate
species are taken from measurements (Andreae and Merlet, 2001; Andreae and
Rosenfeld, 2008; Akagi et al., 2011). Daily fire emissions for 2002-2012 are available
at http://bai.acom.ucar.edu/Data/fire/. In GC-YIBs, all biomass burning emissions are
emitted into the atmospheric boundary layer. The FINN inventory has been widely
used in regional and global chemical transport models (e.g., WRF-Chem and
GEOS-Chem) to quantify the impacts of fires on air quality and weather (Jiang et al.,
2012; Nuryanto, 2015; Vongruang et al., 2017; Brey et al., 2018; Watson et al., 2019).

**2.4 Site-level measurements**


Measurements of surface [$O_3$] in the U.S. are provided by Air Quality System (AQS,
https://www.epa.gov/aqs), those over Europe are provided by European Monitoring
and Evaluation Programme (EMEP, https://emep.int). The observed [$O_3$] at Manaus,
Tg Malim, and Welgegund sites are from earlier studies (Ahamad et al., 2014; Laban
et al., 2018; Pope et al., 2020).

**2.5 Model simulations**


In this study, eight simulations (Table 1) are performed to examine both the direct and
indirect contributions of fires to surface $O_3$. These simulations can be divided into two
main groups:



1. CTRL_FIRE and CTRL_NOFIRE are the control runs using the same emissions

except that the latter omits fire emissions. These runs calculate and output offline

$O_3$ damage, which decreases instantaneous leaf photosynthesis but does not feed

back to affect plant growth and $O_3$ dry deposition.

2. O3CPL_FIRE and O3CPL_NOFIRE are the sensitive experiments that consider

online coupling between $O_3$ and vegetation. These runs include online $O_3$ damage

to plant photosynthesis, which feeds back to affect both vegetation and air

pollution. The two simulations apply the same emissions, except that the latter

omits fire emissions.


For each of these four configurations, two runs are conducted with either high (HS) or
low (LS) $O_3$ damaging sensitivities. All simulations are performed from 2002-2012
using the GC-YIBs model driven by MERRA2 meteorological fields. The first 3 years
are used as spin up, and the results of the last 8 years are analyzed. For the same
configurations, the results from low and high $O_3$ damaging sensitivities are averaged.
The differences between CTRL_NOFIRE and O3CPL_NOFIRE represent the surface
$O_3$ enhancements through $O_3$-vegetation feedback without fire emissions. The
differences between CTRL_FIRE and CTRL_NOFIRE, named O3OFF, represent the
direct contributions of fires to surface $O_3$. The differences between O3CPL_FIRE and
O3CPL_NOFIRE, named O3CPL, represent both direct and indirect contributions of
fires to surface $O_3$. The differences between O3CPL and O3OFF represent the indirect
contributions of fires to surface $O_3$ through $O_3$-vegetation interactions.




## 3 Results

### 3.1 Model validation

Simulated surface daily maximum 8-hour average $O_3$ concentrations (MDA8 [$O_3$],

short for [$O_3$] hereafter) are evaluated using measurements from the AQS and EMEP

datasets over the period of 2005-2012 (Fig 2). The model well captures the observed

spatial distribution of annual [$O_3$] in the U.S. and Europe, with a high correlation

coefficient of 0.51 (p<0.01). Although GC-YIBs overestimates the [$O_3$] in the eastern

U.S. while underestimating it in western Europe, the normalized mean bias (NMB) is

only 4.0%, with a root mean square error (RMSE) of 5.4 ppbv. Therefore, the

simulated $O_3$ vegetation damage in our study is slightly overestimated in the eastern

U.S. but underestimated in western Europe.

254

### 3.2 Direct contributions of fires to $O_3$

Without fire emissions, the simulated global mean annual [$O_3$] is 23.9 ppbv, with a

grid maximum of 63.7 ppbv over the Beijing–Tianjin–Hebei region averaged for

2005-2012 (Fig. 3a). Most high [$O_3$] is distributed in the Northern Hemisphere, where

anthropogenic emissions make the dominant contributions. The inclusion of fire

emissions increases global annual [$O_3$] by an average of 1.2 ppbv (5.0%). Regionally,

the largest enhancement of [$O_3$] by 5.9 ppbv (24.4%) is averaged over central Africa,

with smaller enhancements of 5.7 ppbv (38.2%) averaged over the Amazon, and 3.8

ppbv (10.2%) averaged over southern Asia. Smaller enhancements of 1.1 ppbv (2.2%),



0.9 ppbv (2.1%), and 0.8 ppbv (2.2%) are averaged respectively over eastern China,
western Europe, and the eastern U.S. (Fig. 3b). The predicted fire-induced
enhancements in $[O_3]$ agree well with the simulations using the same model but with
fire emissions from the Global Fire Emission Database (GFED) version 3 (Yue and
Unger, 2018).

We further evaluated the model performance in simulating fire-induced $\Delta[O_3]$ at three
sites across biomass burning regions (Fig. S1). Without fire emissions, the $[O_3]$ is
obviously underestimated, with NMBs of -25.5% at Tg Malim, -53.6% at Manaus,
and -21.3% at Welgegund. As a comparison, simulations with fire emissions show
NMBs in fire seasons of -8.7% at Tg Malim, -1.4% at Manaus, and -15.1% at
Welgegund, suggesting improved $O_3$ simulations by including fire emissions.

**3.3 Fire-induced $O_3$ damages to GPP**
Surface $O_3$ causes strong damage to ecosystem productivity (Fig. 4). Without fire
emissions, surface $O_3$ reduces global annual GPP by 1.7% (3899.8 Tg C $yr^{-1}$, Figs. 4a
and 4c). Regional maximum reductions of 10.9% (372.0 Tg C $yr^{-1}$), 6.1% (366.1 Tg C
$yr^{-1}$), and 4.9% (323.8 Tg C $yr^{-1}$) are averaged respectively over eastern China, the
eastern U.S., and western Europe; these reductions are attributed to the high ambient
$[O_3]$ level and the large stomatal conductance over these regions. The patterns of
$O_3$-induced GPP reductions agree with previous estimates using different models
(Sitch et al., 2007; Yue and Unger, 2015). The inclusion of fire emissions causes





additional GPP reductions. Globally, fire-induced $\Delta O_3$ decreases annual GPP by 0.4%
(1312.0 Tg C yr$^{-1}$, Figs. 4b and 4d). Regionally, the largest GPP reduction of 1.4%
(370.3 Tg C yr$^{-1}$) is averaged over the Amazon due to the largest enhancement of [$O_3$]
caused by fires. Furthermore, fire $\Delta$[$O_3$] causes additional annual GPP reductions of
1.3% (358.0 Tg C yr$^{-1}$), averaged over central Africa, and 1.0% (77.1 Tg C yr$^{-1}$),
averaged over southern Asia. In contrast, limited damage is found in eastern China,
western Europe, and the eastern U.S. due to low fire $\Delta$[$O_3$]. Following the changes in
GPP, fire-induced $O_3$ damage to LAI shows a regional maximum of 0.3-0.7% in
central Africa and a global reduction of 0.02-0.5% (Fig. S2).

**3.4 Indirect contributions of fires to $O_3$**
Vegetation parameters such as LAI and stomatal conductance play important roles in
modulating surface [$O_3$]. The $O_3$-induced changes in these variables interactively feed
back to alter local [$O_3$] (Fig. 5). Without fire emissions, the annual $\Delta$[$O_3$] from
$O_3$-vegetation interactions is limited to eastern China by 0.5 ppbv, the eastern U.S. by
0.3 ppbv, and western Europe by 0.2 ppbv. The largest grid positive feedback of up to
0.8 ppbv is found in the eastern U.S. (Figs. 5a and 5c). Sensitivity experiments further
show that such enhancement of surface [$O_3$] mainly results from the inhibition of
stomatal conductance by $O_3$ stomatal uptake (Fig. S3a), which reduces the $O_3$ dry
deposition velocity (Fig. S4). Consequently, large $\Delta$[$O_3$] (Figs. 5a and 5c) are
collocated with areas enduring high levels of $O_3$ vegetation damage (Figs. 4a and 4c).
As a comparison, the feedback of LAI changes is generally small (Fig. S3b), which is



mainly attributed to limited $O_3$ damage on LAI (Fig. S2). The enhancement of $[O_3]$
from fires causes additional feedback to the surface $[O_3]$. The largest annual $\Delta[O_3]$ of
0.13 ppbv due to $O_3$-vegetation feedback is averaged on over the Amazon (Figs. 5b
and 5d), where the highest GPP reductions by fire-induced $O_3$ are predicted (Figs. 4b
and 4d). Such feedback additionally enhances local $[O_3]$ by 0.12 ppbv, averaged over
central Africa, and 0.09 ppbv, averaged over southern Asia. However, limited
$O_3$-vegetation feedback is found in the eastern U.S., eastern China, and western
Europe, either because of low fire-induced $\Delta[O_3]$ (Fig. 3b) or low $\Delta$GPP (Figs. 4b and
4d). The changes in $O_3$ dry deposition velocity broadly match the pattern of
$O_3$-vegetation feedback (Fig. S4), suggesting that reduced dry deposition velocity due
to $O_3$-induced inhibition of stomatal conductance is the dominant driver for the
enhanced surface $[O_3]$.

Fig. 6 shows seasonal variations in $O_3$-vegetation feedback. Without fire emissions,
$O_3$-vegetation feedback in eastern China, the eastern U.S., and western Europe shows
similar seasonal variations, increasing from January to July and then decreasing (Fig.
6a). For these regions, surface $[O_3]$ and stomatal conductance reach maximums during
the growth season (May-October), resulting in instantaneous $O_3$ uptake. Therefore,
$O_3$-vegetation interactions are expected to be stronger during the growth season in the
Northern Hemisphere. However, $O_3$-vegetation feedback driven by fires in the
Amazon and Southern Asia reaches a maximum during August-December and
February-June, respectively. Moreover, double peaks are shown in central Africa, with





maximums during February-April and July-September (Fig. 6b). The distinct seasonal
variations in biomass burning regions are attributed to fire emissions. At low latitudes,
stomatal conductance shows limited seasonal variations. Therefore, $O_3$-vegetation
feedback driven by fires is mainly dependent on fire-induced $\Delta[O_3]$.

Fire-induced $O_3$ shows stronger interactions with vegetation under favorable
meteorological conditions. We sort daily $\Delta[O_3]$ from $O_3$-vegetation feedback and
calculate the average of $\Delta[O_3]$ above the 95th percentile (Fig. S5). The spatial pattern
of $\Delta[O_3]$ during extreme $O_3$-vegetation feedback is broadly consistent with that of the
annual average, albeit with much stronger $O_3$-vegetation feedback. Without fire
emissions, $O_3$-vegetation feedback enhances $[O_3]$ by 2.0 ppbv averaged over eastern
China, 1.8 ppbv averaged over the eastern U.S., and 1.1 ppbv averaged over western
Europe (Figs. S5a and S5c). Fire emissions alone enhance $[O_3]$ through $O_3$-vegetation
interactions by 1.1 ppbv averaged over the Amazon, 0.8 ppbv averaged over southern
Asia, and 0.6 ppbv averaged over central Africa during extreme $O_3$-vegetation
feedback (Figs. S5b and S5d).

**3.5 Indirect vs. direct contributions of fires to $O_3$**
We further compare the indirect and direct contributions of fire emissions to surface
$[O_3]$. Here, the direct contributions indicate $\Delta[O_3]$ caused by fire emissions of
chemical precursors, while the indirect contributions represent additional $\Delta[O_3]$ from
$O_3$-vegetation interactions caused by fire-induced $O_3$. Without fire emissions,



O$_3$-vegetation interactions cause enhancement of [O$_3$] by 1.0% averaged over eastern
China, 0.8% averaged over the eastern U.S., and 0.5% averaged over western Europe
(Figs. 7a and 7c). Compared to nonfire sources, fire emissions cause larger
perturbations in surface [O$_3$] through O$_3$-vegetation interactions (Figs. 7b and 7d). The
ratios of indirect to direct annual $\Delta$[O$_3$] are 3.7% averaged over eastern China, 2.0%
averaged over the eastern U.S., and 1.6% averaged over western Europe. For these
regions, the absolute $\Delta$[O$_3$] from direct fire emissions is usually lower than 1 ppbv
(Fig. 3b). However, the high level of ambient [O$_3$] (Fig. 3a) provides a sensitive
environment in which moderate increases in [O$_3$] from fires can cause large indirect
contributions to regional [O$_3$] through vegetation damage. For fire-prone regions, the
ratios of indirect to direct annual $\Delta$[O$_3$] are 2.6% averaged over southern Asia, 1.9%
averaged over the eastern U.S., and 1.4% averaged over central Africa.

**3.6 Aggravated O$_3$ damage to GPP through O$_3$-vegetation feedback**
The additional O$_3$ enhancement can exacerbate the damaging effects on vegetation.
Without fire emissions, online O$_3$ causes a global annual GPP reduction of 0.2%
(299.6 Tg C yr$^{-1}$, Figs. S6a and S6c) from the offline O$_3$. Regionally, additional
reductions are mainly found in eastern China, the eastern U.S., and western Europe,
where GPP is further decreased by 27.1 Tg C yr$^{-1}$, 40.8 Tg C yr$^{-1}$ and 28.4 Tg C yr$^{-1}$,
respectively. For fire emissions, the online fire-induced $\Delta$O$_3$ results in a higher GPP
reduction by 25.0 Tg C yr$^{-1}$ averaged over the Amazon, and 24.3 Tg C yr$^{-1}$ averaged
over central Africa, and 7.1 Tg C yr$^{-1}$ averaged over southern Asia compared to the



offline fire-induced $\Delta O_3$ (Figs. S6b and S6d). Such spatial patterns are broadly
consistent with $\Delta[O_3]$ induced by $O_3$-vegetation feedback (Fig. 5).

**4 Conclusions and discussion**
Many studies have explored the direct contributions to surface $O_3$ by fire emissions.
However, the feedback of fire-induced $O_3$ vegetation damage to surface $[O_3]$ remains
unquantified. In this study, we find that fire-induced $O_3$ causes a positive feedback to
surface $[O_3]$ mainly because of the inhibition effects on stomatal conductance.
Regionally, $O_3$-vegetation feedback driven by fires enhances surface annual $[O_3]$ by
0.13 ppbv averaged over the Amazon, 0.12 ppbv averaged over central Africa, and
0.09 ppbv averaged over southern Asia. Such feedback exhibit large seasonal
variations, with the maximums of 0.5 ppbv averaged over the Amazon in October, 0.3
ppbv averaged over southern Asia in April, and 0.2 ppbv averaged over central Africa
in April. During extreme $O_3$-vegetation interactions, the feedback can rise to >0.6
ppbv in these fire-prone areas. Although direct formations of $O_3$ from fires are limited
in eastern China and the eastern U.S., the feedback of $O_3$-vegetation coupling results
in additional enhancement of surface $[O_3]$ by 3.7% and 2.0% upon the fire-induced
$\Delta[O_3]$. Such large ratios in these regions are attributed to the high level of ambient $[O_3]$
that provides a sensitive environment in which moderate increases in $[O_3]$ from fires
can cause large indirect contributions to regional $[O_3]$ through vegetation damage.

Some uncertainties may affect the conclusions of this study. First, we employed a



model resolution of 4°×5° due to the limitations in computational resources. We
performed a one-year sensitivity simulation at a 2°×2.5° resolution. The comparisons
show that fire-induced direct $O_3$ enhancement is very similar between the simulations
with low and high resolutions, although the former runs predict slightly higher
changes in $[O_3]$ than the latter (Fig. S7). Second, different biomass burning datasets
may affect the estimated $O_3$-vegetation feedback in our study. At present, the
FINNv1.5 and GFEDv4.1 inventories are available in the public-release of
GEOS-Chem v12.0.0. Compared with the FINNv1.5 inventory, simulations using the
GFEDv4.1 inventory predict a lower $O_3$-vegetation feedback in the Amazon (Fig. S8a)
and southern Asia (Fig. S8c) but a higher $O_3$-vegetation feedback in central Africa
(Fig. S8b). Finally, fires can decrease VOC emissions from biogenic sources by
burning vegetation. However, compared to the VOCs emitted by fires, the VOC loss
from burned vegetation is generally smaller (Fig. S9). Therefore, the influence of
reduced VOCs from vegetation loss on surface $[O_3]$ can be ignored.

Despite these uncertainties, we present the first estimate of $O_3$ enhancement by fire
emissions through $O_3$-vegetation interactions. Such enhancement is not limited to
fire-prone regions, but is also significant over downwind areas with high ambient $[O_3]$
levels. Although the absolute perturbations may be moderate for the whole fire season,
$O_3$-vegetation interactions can largely increase surface $O_3$ during extreme
$O_3$-vegetation interactions, leading to additional threats to public health and
ecosystem productivity.




**Data availability**

The site-level [O$_3$] in the U.S. can be download from AQS (https://www.epa.gov/aqs).
The site-level [O$_3$] in the Europe can be download from EMEP (https://emep.int). The
observed [O$_3$] at Manaus, Tg Malim, and Welgegund sites are from earlier studies
(Ahamad et al., 2014; Laban et al., 2018; Pope et al., 2020). The GC-YIBs simulation
results are available from the corresponding authors on request.

**Competing interests.** The authors declare no competing financial interests.

**Author Contributions.** XY conceived the study. YL conducted the model
simulations. YL and XY were responsible for results analysis. HL, LZ, and YY
revised and improved the manuscript. HZ, CT, and CG helped prepare model input.
YM, LG, and YC helped prepare observation dataset.

**Acknowledgements.** This work was supported by Jiangsu Science Fund for
Distinguished Young Scholars (grant no. BK20200040), the National Natural Science
Foundation of China (grant no. 41975155), and the National Key Research and
Development Program of China (grant nos. 2019YFA0606802 and
2017YFA0603802).




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






**Table 1** Summary of simulations using the GC-YIBs model

| Name | Emissions | O₃ damaging | O₃ sensitivities |
|---|---|---|---|
| CTRL_FIRE_HS | All including fires | Offline | High |
| CTRL_FIRE_LS | All including fires | Offline | Low |
| CTRL_NOFIRE_HS | All but without fires | Offline | High |
| CTRL_NOFIRE_LS | All but without fires | Offline | Low |
| O3CPL_FIRE_HS | All including fires | Online | High |
| O3CPL_FIRE_LS | All including fires | Online | Low |
| O3CPL_NOFIRE_HS | All but without fires | Online | High |
| O3CPL_NOFIRE_LS | All but without fires | Online | Low |















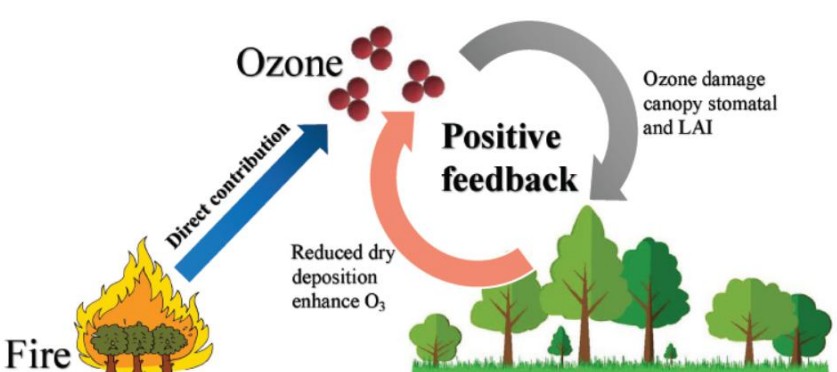


**Figure 1** Diagram of the impacts of fires on surface $O_3$ through direct emissions and
$O_3$-vegetation feedback.


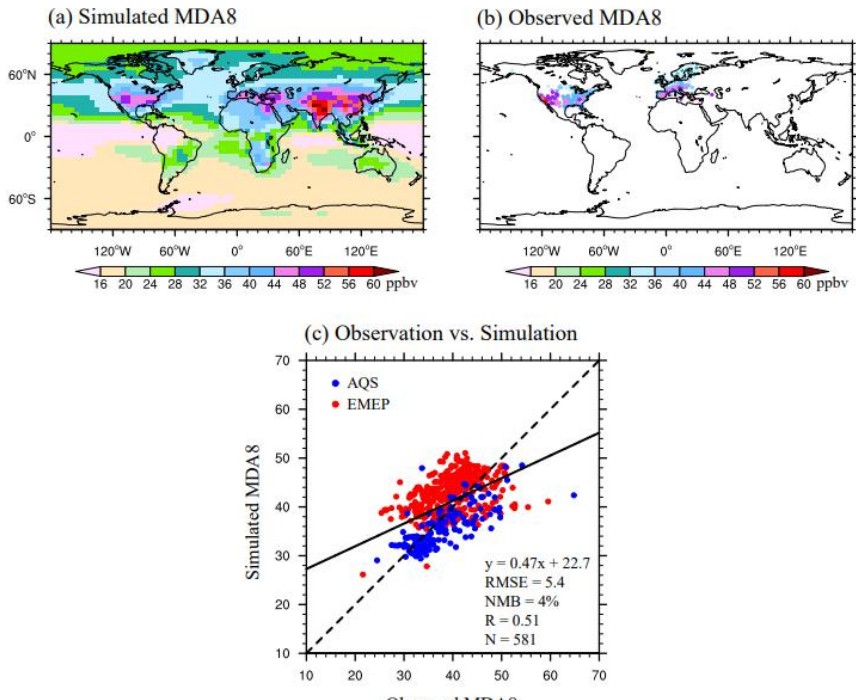

**Figure 2** Spatial pattern of (a) simulated and (b) observed surface [O₃]. (c) Scatter

plot of surface [O₃] over measurements in two regions. The black line shows the

linear regression between the observed and simulated [O₃]. The regression fit,

correlation coefficient (R), root mean square error (RMSE), and normalized mean bias

(NMB) are shown in the bottom panel with an indication of site numbers (N) used for

statistics.


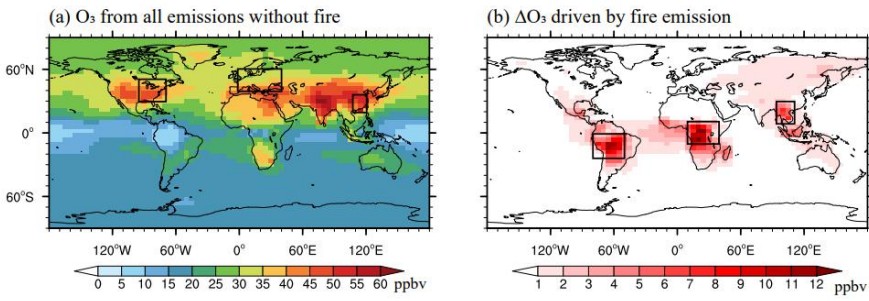

**Figure 3** Annual surface [O₃] from (a) nonfire and (b) fire-alone sources. The six subregions are marked with black boxes: Eastern U.S. (EUS, 30°N-50°N, 95°W-70°W), Western Europe (WEU, 40°N-60°N, 0°-40°E), Eastern China (ECH, 20°N-35°N, 108°E-120°E), Amazon (AMZ, 25°S-0°, 80°W-50°W), Central Africa (CAF, 10°S-10°N, 10°E-40°E), and Southern Asia (SAS, 10°N-30°N, 95°E-110°E).





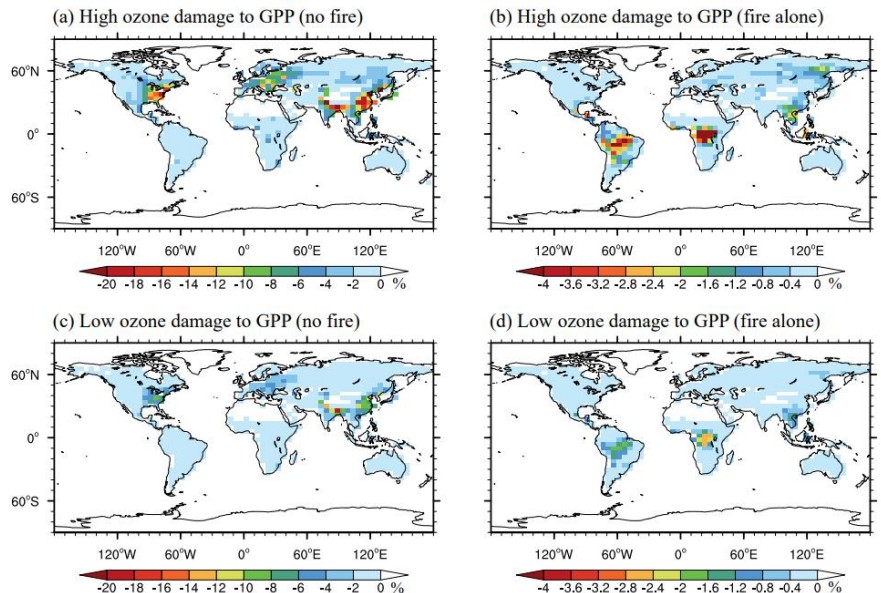

**Figure 4** Annual percentage of reductions in GPP caused by $O_3$ from (a, c) nonfire

and (b, d) fire alone sources with (a, b) high and (c, d) low $O_3$ sensitivities. Please

note the differences in color scales.



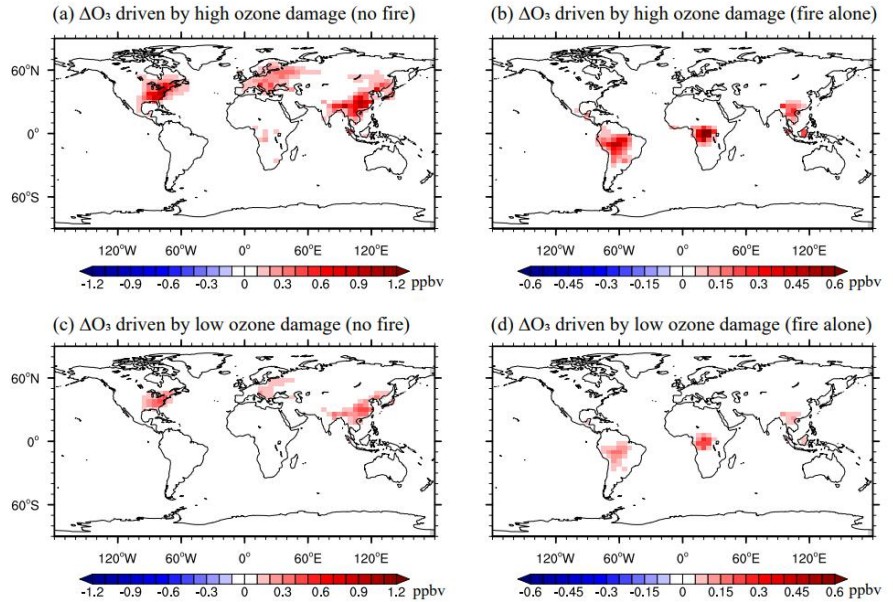

717

**Figure 5** Annual feedback to surface O₃ caused by O₃ vegetation damage with (a, b)

high and (c, d) low O₃ sensitivities. (a) and (c) represent feedback by O₃ from nonfire

sources; (b) and (d) represent feedback by O₃ from fire emissions alone. Please note

the differences in color scales.




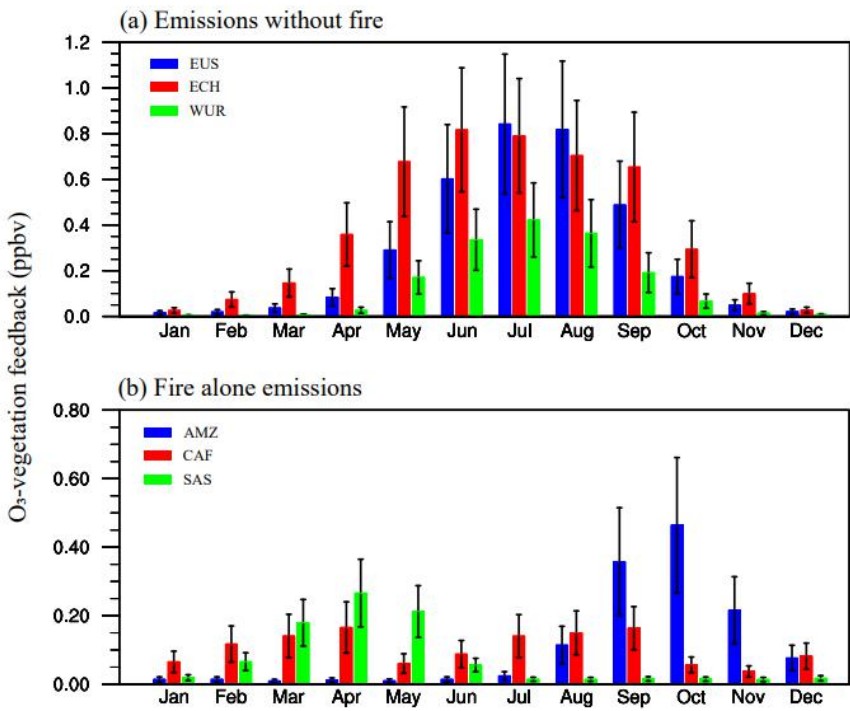


**Figure 6** Seasonal variations in O₃-vegetation feedback driven by (a) nonfire and (b)

fire-alone sources. The error bars represent low to high O₃ damaging sensitivities.














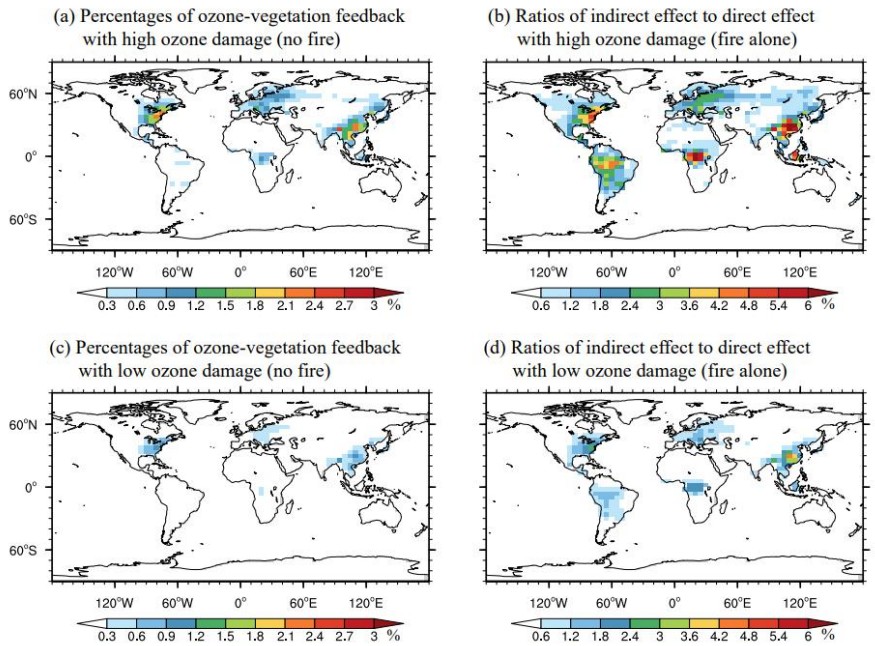

**Figure 7** Annal ratios of indirect $\Delta[O_3]$ to ambient $[O_3]$ from (a, c) nonfire emissions

and the ratios of indirect to direct $\Delta[O_3]$ from (b, d) fire emissions alone with (a, b)

high and (c, d) low $O_3$ damaging sensitivities. Please note the differences in color

scales.