# Peer review of "Indirect contributions of global fires to surface ozone through ozone-vegetation feedback"

_Atmospheric Chemistry and Physics, 2020_

## Author Comment (AC1)

We are grateful to the referee for his/her time and energy in providing helpful comments and guidance that have improved the manuscript. In this document, we describe how we have addressed the reviewer's comments. Referee comments are shown in black and author responses are shown in blue text.

**Reviewer#1**

Well written paper showing vegetation feedback of fire-enhanced $O_3$ based on modelling approaches. The results are supported by showing $O_3$ vegetation feedback with and without fire. I would recommend to slightly change the Introduction so that the reader would be able to follow the text more fluently and the paragraphs will follow each other more logically.

➔ Thank you for your positive evaluations. All the questions and concerns have been carefully answered. We have adjusted the sequence of paragraphs in the Introduction section, so that the readers can follow the text more fluently.

Here are some more detailed comments:

1. line 23: fire is not a source of ozone. Please rewrite it in a manner that it produces precursors of $O_3$. Moreover, the sentence has double meaning - it is a fire and $O_3$, which causes damage to vegetation and reduces stomatal conductance.

*Response*: We revised this sentence to avoid confusions: "Fire is an important source of ozone ($O_3$) precursors. The formation of surface $O_3$ can cause damages to vegetation and reduce stomatal conductance." (Lines 23-24)

2. line 70-71: here you write the same as in line 49-50.

*Response*: The original sentence "Emissions from biomass burning generate a large amount of $O_3$ precursors" on lines 70-71 has been removed in the revised paper.

3. line 79: cite the three papers here.

*Response*: We added a recent paper by Zhu et al. (2021) in the revised paper. All the

four studies have been cited as suggested. (Lines 67-68)

4. line 88-90: when the $O_3$ is enhanced, one would expect higher deposition velocity. Could you explain why in that case it is the opposite?

*Response*: Vegetation acts as a major sink through stomatal uptake of $O_3$. Observations show that surface $O_3$ damages will decrease both plant photosynthesis and stomatal conductance. Therefore, the higher surface $O_3$ results in lower stomatal conductance, leading to smaller dry deposition velocity. Such $O_3$-vegetation feedback has been revealed in previous studies and is further clarified as follows: "Simulations showed that surface $O_3$ could be enhanced … through comparable effects from biogeochemical (decreased dry deposition and increased isoprene emissions) and biogeophysical (changes in meteorological variables following reduced transpiration rate) feedbacks from $O_3$-vegetation interactions." (Lines 70-74)

5. line 103-105: how do the influence the sources and sinks? I think it influence more sinks. Now it seems as the reduces LAI would be a source of $O_3$, when it is just reducing sink of $O_3$. Moreover, there is a new review about $O_3$ effect on vegetation, which you might consider to include here. 10.3390/atmos12010082

*Response*: Vegetation stomatal conductance and LAI influence dry deposition of $O_3$. LAI also influences the emissions of BVOC, which is an important $O_3$ precursor. The new review article has been cited in lines 46-47. "Tropospheric ozone ($O_3$) is a toxic air pollutant with detrimental effects on vegetation (Yue and Unger, 2014; Juráň et al., 2021)"

6. line 203-204: is that true? One would expect oxidation of the compounds and sedimentation of particles before reaching PBL

*Response*: We agree with your comment. Most of fire emissions and oxidations occur below the PBL. We clarified as follows: "In GC-YIBs, all biomass burning emissions occur in the atmospheric boundary layer. Such configuration might slightly overestimate regional $O_3$ formation as observations suggested ~20% of fire plumes

reached the height above the boundary layer (Val Martin et al., 2010) and consequently enhanced surface $O_3$ level at the downwind regions (Jaffe and Wigder, 2012)." (Lines 214-219)

7. line 256-258: is the mean annual or is the mean from 2005-2012, which is longer time that annual.

Response: It is the mean [$O_3$] during 2005-2012. "global mean annual" has been modified as "global mean" in the revised paper. (Lines 274-275)

8. line 381: I think you just hypothesize, that it is due to reduced stomatal conductance. There is no model feedback showing this nor your measurement.

*Response*: This conclusion is not a hypothesis. First, many observations have shown that $O_3$ can cause damages to stomatal conductance (Yue et al., 2016). Second, our simulations show that $O_3$-induced reductions in stomatal conductance result in enhanced surface $O_3$ (Fig. S3a) due to reduced dry deposition velocity (Fig. S4) (Related descriptions in Lines 326-332). Third, such $O_3$-vegetation feedback has been revealed and supported by other modeling studies (Sadiq et al., 2017; Zhou et al., 2018; Gong et al., 2020; Zhu et al., 2021).

9. Fig 6: explain abbreviations AMZ, CAF, SAS as in Fig. 3

*Response*: We modified the figure caption as follows: "The blue, red, and green bars in (a) represent the $O_3$-vegetation feedback in Eastern U.S. (EUS), Eastern China (ECH), Western Europe (WUR), respectively. The blue, red, and green bars in (b) represent the $O_3$-vegetation feedback in Amazon (AMZ), Central Africa (CAF), and Southern Asia (SAS), respectively." (Lines 718-721)

**References**

Gong, C., Lei, Y., Ma, Y., et al. Ozone-vegetation feedback through dry deposition and isoprene emissions in a global chemistry-carbon-climate model. Atmospheric Chemistry and Physics[J]: 2020, 20: 3841-3857.

Jaffe, D. A. and Wigder, N. L. Ozone production from wildfires: A critical review. Atmospheric Environment[J]: 2012, 51: 1-10.

Juráň, S.,Grace, J. and Urban, O. Temporal Changes in Ozone Concentrations and Their Impact on Vegetation. Atmosphere[J]: 2021, 12: 82.

Sadiq, M., Tai, A. P. K., Lombardozzi, D., et al. Effects of ozone-vegetation coupling on surface ozone air quality via biogeochemical and meteorological feedbacks. Atmospheric Chemistry and Physics[J]: 2017, 17: 3055-3066.

Val Martin, M., Logan, J. A., Kahn, R. A., et al. Smoke injection heights from fires in North America: analysis of 5 years of satellite observations. Atmospheric Chemistry and Physics[J]: 2010, 10: 1491-1510.

Yue, X., Keenan, T. F., Munger, W., et al. Limited effect of ozone reductions on the 20-year photosynthesis trend at Harvard forest. Global Change Biology[J]: 2016, 22: 3750-3759.

Yue, X. and Unger, N. Ozone vegetation damage effects on gross primary productivity in the United States. Atmospheric Chemistry and Physics[J]: 2014, 14: 9137-9153.

Zhou, S. S., Tai, A. P. K., Sun, S. H., et al. Coupling between surface ozone and leaf area index in a chemical transport model: strength of feedback and implications for ozone air quality and vegetation health. Atmospheric Chemistry and Physics[J]: 2018, 18: 14133-14148.

Zhu, J.,Tai, A. P. K. and Yim, S. H. L. Effects of ozone-vegetation interactions on meteorology and air quality in China using a two-way coupled land-atmosphere model. Atmos. Chem. Phys. Discuss. [preprint][J]: 2021, doi: https://doi.org/10.5194/acp-2021-165, 2021.

---

## Author Comment (AC2)

We are grateful to the referee for his/her time and energy in providing helpful comments and guidance that have improved the manuscript. In this document, we describe how we have addressed the reviewer's comments. Referee comments are shown in black and author responses are shown in blue text.

**Reviewer#2**

This paper examines the effects of fires on surface ozone pollution and the subsequent feedback effects that may further enhance ozone. This runs along the excellent of work this group of researchers have done demonstrating the importance of ozone-vegetation interactions in atmospheric chemistry modeling and air quality projections. While the idea of ozone-vegetation feedbacks is not new by now, this paper presents a new perspective by focusing on fires, which distinguishes itself from previous ozone-vegetation papers that focused on anthropogenic emissions. There are however several aspects which I believe need to be addressed, and revisions need to be made, before this paper can be published. Please see below for my comments and suggestions.

Thank you for your positive evaluations. All the questions and concerns have been carefully answered and the paper has been revised accordingly.

1. P4 L68-74: First of all, it should be "vegetation damage" that would influence the sources and sinks of ozone via various "feedbacks". Second, the authors mentioned the distinction between "biogeochemical" and "biogeophysical" feedbacks, but it needs to be explained further. What are the distinctions? In particular, in the following few sentences, only "biogeochemical" processes are considered, but the "biogeophysical" pathways are not mentioned at all. In general, the whole introduction lacks a thorough illustration of the detailed feedback pathways and the distinctions between the biogeochemical and biogeophysical effects of vegetation on air quality (and thus feedbacks after ozone damage). I suggest having a separate paragraph detailing first how vegetation processes affect ozone air quality,

distinguishing between the biogeochemical (i.e., BVOC emissions and dry deposition) and biogeophysical (i.e., transpiration and the subsequent changes in meteorological environment) pathways. A paper that can be referenced on these is Wang et al. (2020). After such an introduction, the feedback effects can be explained much more clearly. *Response*: In the revised paper, we clarify as follows: "In turn, vegetation damage also influences both the sources and sinks of O3 through biogeochemical and biogeophysical feedbacks (Curci et al., 2009; Heald and Geddes, 2016; Fitzky et al., 2019). The damaged vegetation decreases isoprene emissions and stomatal conductance (Wittig et al., 2009; Feng et al., 2019), which influence O3 production and dry deposition. Moreover, weakened leaf-level transpiration following O3 damage modulates meteorological parameters, such as surface air temperature and atmospheric relative humidity, leading to substantial biogeophysical feedbacks on surface O3 (Lombardozzi et al., 2012; Sadiq et al., 2017)". (Lines 54-62)

2. P5 L89-90: In Sadiq et al. (2017), much of the positive feedback is due to "biogeophysical" effects, i.e., reduced transpiration leading to higher surface temperature and thus higher isoprene emissions, then higher ozone. Reduced dry deposition velocity is roughly only half of explanation. In general, in this whole paragraph, the distinctions in methodology or pathways included should be explained more clearly. E.g., Zhou et al. (2018) and Gong et al. (2020) only considered biogeochemical effects, because in their models, climate was not dynamically simulated, whereas Sadiq et al. (2017) considered both effects because their model dynamically simulated climate. Moreover, a fourth study (Zhu et al., 2021) that focused on China is currently under review.

*Response*: Thank you for your suggestions. We would like to explain that Gong et al., (2020) applied a chemistry-carbon-climate coupled model and considered both biogeochemical and biogeophysical feedbacks. In the revised paper, we have modified this part as follows: "At present, the feedbacks from  $O_3$ -damaging vegetation on  $O_3$  have only been examined by four papers (Sadiq et al., 2017; Zhou et al., 2018; Gong et al., 2020; Zhu et al., 2021). Sadiq et al. (2017) implemented a

parameterization of O3 vegetation damage into a climate model and quantified online O3-vegetation coupling. Simulations showed that surface O3 could be enhanced by up to 4-6 ppbv over Europe, North America, and China through comparable effects from biogeochemical (decreased dry deposition and increased isoprene emissions) and biogeophysical (changes in meteorological variables following reduced transpiration rate) feedbacks from O3-vegetation interactions. Similar conclusions were achieved by Zhu et al. (2021), who investigated the effects of O3-vegetation interaction in China using a two-way coupled land-atmosphere model. By including O3 damage to isoprene emissions in a fully coupled global chemistry-carbon-climate model, Gong et al. (2020) highlighted that such O3-vegetation positive feedbacks were mainly driven by reduced dry deposition following O3 damage to photosynthesis. Different from above three studies, Zhou et al. (2018) implemented steady-state O3-induced LAI changes into GEOS-Chem and quantified only the influences of O3-vegetation biogeochemical feedbacks because the model is driven with prescribed meteorological fields. Results showed that O3-induced damage to LAI can enhance O3 by up to 3 ppbv in the tropics, eastern North America, and southern China through changes in dry deposition and isoprene emissions." (Lines 66-85)

3. P5 L100: A better justification is needed here to illustrate why this is important to look at. It's unquantified, but do we really expect the ozone-vegetation feedback via fires is really gone be important? Any justification for this expectation (and thus the motivation of this paper)? Any comparison with previous work regarding the magnitude of the potential effects?

*Response*: In the revised paper, we clarified as follows: "Regionally, especially in Amazon and central Africa, fires can enhance surface  $O_3$  by 10-30 ppbv through emissions of NOx and VOCs during fire seasons (Yue and Unger, 2018; Pope et al., 2020). Over these regions, strong O3-vegetation interactions are expected because of high fire O3 concentrations and dense vegetation cover. Previous studies showed that fire O3 causes large GPP reduction of 200-400 Tg C yr-1 over Amazon and central Africa (Pacifico et al., 2015; Yue and Unger, 2018). With likely increased wildfire

activity due to global warming, surface  $O_3$  will be further enhanced by wildfires (Amiro et al., 2009; Balshi et al., 2009; Wang et al., 2016; Yue et al., 2017), leading to more severe  $O_3$  damage on vegetation. Although the feedback of vegetation damage on surface  $O_3$  have been well explored on global (Sadiq et al., 2017; Zhou et al., 2018; Gong et al., 2020) or regional (Zhu et al., 2021) scales, these studies all focused on  $O_3$ -vegetation from combined anthropogenic and natural sources. Therefore, quantification of the  $O_3$ -vegetation interactions associated with fire emissions is very important for a comprehensive understanding of the effects of fires on surface  $O_3$ ". (Lines 96-110)

4. P8 L160: The setting of this model using prescribed meteorology needs to be emphasized and contrasted with fully coupled climate-chemistry-vegetation models such as CESM. It should also be emphasized that this model setting only addresses the "biogeochemical" effects, not "biogeophysical" (referring to the points made above).

*Response*: The related descriptions "It should be noted that only biogeochemical feedbacks from  $O_3$  vegetation damage on surface  $O_3$  are considered in this study because GC-YIBs uses prescribed meteorology (MERRA2)" have been added in the revised paper. (Lines 256-259)

5. P9 L181: Does YIBs actually simulate a multi-layer canopy, instead of a big-leaf canopy? This needs to be clarified. If a multi-layer canopy is represented, the number of layers and other canopy parameter setting needs to be clarified. If not, this line here should be corrected.

*Response*: A multi-layer canopy has been applied in the YIBs model. The related descriptions "The canopy is divided into an adaptive number of layers (typically 2-16) for light stratification. The YIBs model applies a well-established Michaelis–Menten enzyme kinetics scheme to compute the leaf photosynthesis (Farquhar et al., 1980; Von Caemmerer and Farquhar, 1981), which is further upscaled to the canopy level by the separation of sunlit and shaded leaves (Spitters, 1986)" have been added in the

6. P11 L230-241: An obviously missing element in their model setting and experiments is that fires also damage LAI and canopy height directly, which may only happen only where fires happen but would be the dominant effect (other than ozone damage on plants) there. Fires also influence the long-term recovery and growth of the forests, which of course would also influence ozone. I understand that such an effect is more localized to the forested areas while the ozone-vegetation feedbacks can occur downwind of the forests, the lack of consideration of this necessary pathway should be explained upfront early on. Indeed, this should also be discussed as early as in the introduction.

*Response*: In the revised paper, we acknowledge this limitation in discussion section: "(i) The GC-YIBs simulations do not consider the direct fire damages to vegetation and the consequent long-term recovery of forests. In our study, we focus only on the feedbacks of fire-induced O3-vegetation interactions to surface O3". (Lines 419-422)

7. P13 L285: It should be clarified that the reductions are consistent with studies/models that used the same ozone damage scheme. It should also be mentioned that some other studies, using other ozone damage scheme, e.g., the Lombardozzi scheme (Zhou et al., 2018; Zhu et al., 2021), may find quite different ozone-induced reductions in GPP.

*Response*: In the revised paper, we clarified as follows: "The patterns of  $O_3$ -induced GPP reductions agree with previous estimates using the same  $O_3$  damage schemes (Sitch et al., 2007; Yue and Unger, 2015). However, compared to simulations using another scheme (Lombardozzi et al., 2012; Zhou et al., 2018; Zhu et al., 2021), this study estimates smaller GPP reductions. Such discrepancy indicates there are large uncertainties in  $O_3$  vegetation damage schemes, and more observations should be developed to evaluate different schemes in future studies". (Lines 301-307)

8. P14 L303: The reduction in stomatal conductance mainly follows reduced

photosynthesis – this should be clarified. This is obviously missing some newer physiology that people have found recently, e.g., the sluggishness of stomatal responses after ozone damage (Huntingford et al., 2018) that may cause the stomata to be more open under ozone exposure than otherwise. Such missing element needs to be discussed.

*Response*: (i) This sentence has been modified as "Sensitivity experiments further show that such enhancement of surface  $[O_3]$  mainly results from the inhibition of stomatal conductance following reduced photosynthesis by  $O_3$  damage (Fig. S3a)". (Lines 326-328)

(ii) We discussed this limitation in discussion section: "(iii) There is evidence that O3 exposure may cause "sluggishness" that delays the stomatal responses to O3 damage (Huntingford et al., 2018). However, we do not include "sluggishness" in our scheme because its net impacts on stomatal conductance remain uncertain. For example, observations found that the increased short-term water loss (delayed stomatal responses) may be offset by the decreased long-term water loss (lower steady-state stomatal conductance) with the stomatal "sluggishness" (Paoletti et al., 2019)". (Lines 426-432)

9. P17 L354: Why does fire emission cause larger ozone-vegetation feedbacks than non-fire sources? It needs to be explained.

*Response:* In this part, we focus on the ratios of indirect to direct. The absolute O3-vegetation feedbacks are smaller in China and U.S. because of the smaller direct contributions from fire on O3 by emitting substantial number of precursors. But we further compared the indirect and direct contributions of fire emissions to surface O3 and found that the largest ratios of indirect to direct  $\Delta$ [O3] are 3.7% in eastern China and 2.0% in eastern U.S.. The explanations are shown in next response.

10. P17 L359-363: The rationale behind needs to be explained in greater detail as well.

Response: As explained in "Scheme of O3 vegetation damage" section, the impacts of

 $O_3$  exposure on photosynthesis and stomatal conductance are dependent on excessive  $O_3$  flux, which is calculated as the difference between stomatal  $O_3$  flux and damaging thresholds. In our simulations, the background  $[O_3]$  is defined as  $[O_3]$  from all sources except for fire emissions. Fire emissions act as a disturbance of background  $[O_3]$ . For anthropogenic regions (eastern U.S. and eastern China), the higher background  $[O_3]$  makes it easier to exceed the damaging thresholds and cause  $O_3$ -vegetation feedback. Therefore, fire emissions will cause a larger ratio of indirect to direct than anthropogenic emissions for same  $[O_3]$  of 1 ppbv.

In the revised paper, we modified as "Compared to nonfire sources, fire emissions cause larger relative perturbations in surface  $[O_3]$  through O3-vegetation interactions (Figs. 7b and 7d). The ratios of indirect to direct annual  $\Delta[O_3]$  are 3.7% averaged over eastern China, 2.0% averaged over the eastern U.S., and 1.6% averaged over western Europe. For these regions, the absolute  $\Delta[O_3]$  from direct fire emissions is usually lower than 1 ppbv (Fig. 3b). However, the high level of background [O\_3] (all sources except fire emissions, Fig. 3a) provides such a sensitive environment that the moderate increases of [O3] from fires can cause large feedback to regional surface [O3] through vegetation damage". (Lines 377-385)

11. P19 L406-409: This is also related to my comments on P11 L230-241 above. Fires do not only affect BVOC by burning vegetation, it also reduces LAI and the long-term recovery and growth of the forests, thus affect the whole ozone-vegetation interactions in the long term. When a forest is burned, the reductions in LAI, dry deposition, transpiration and BVOC emissions can have effects that last for many years, and this temporal perspective is entirely missing from the current discussion. A more thorough discussion on this missing element, and the implications on the validity and significance of this paper's results, is warranted.

*Response*: Thank you for your suggestions. We agree that fires burn forest and cause reductions in LAI, dry deposition, transpiration, and BVOC emissions. However, such perturbations are related to the land use and land cover changes, instead of the

 $O_3$ -vegetation feedback, and are not the main focus of this study. In the revised paper, we added the following discussion: "(i) The GC-YIBs simulations do not consider the direct fire damages to vegetation and the consequent long-term recovery of forests. In our study, we focus only on the feedbacks of fire-induced  $O_3$ -vegetation interactions to surface  $O_3$ . (ii) Fires can decrease VOC emissions from biogenic sources by damaging vegetation directly. However, compared to the VOCs emitted by fires, the VOC loss from burned vegetation is generally smaller (Fig. S7). Therefore, the influence of reduced VOCs from vegetation loss on surface  $[O_3]$  can be ignored". (Lines 419-426)

**References:**

Huntingford, C., Oliver, R. J., Mercado, L. M., and Sitch, S.: Technical note: A simple theoretical model framework to describe plant stomatal "sluggishness" in response to elevated ozone concentrations, Biogeosciences, 15, 5415–5422, https://doi.org/10.5194/bg-15-5415-2018, 2018.

Wang, L., Tai, A. P. K., Tam, C.-Y., Sadiq, M., Wang, P., and Cheung, K. K. W.: Impacts of future land use and land cover change on mid-21st-century surface ozone air quality: distinguishing between the biogeophysical and biogeochemical effects, Atmos. Chem. Phys., 20, 11349–11369, https://doi.org/10.5194/acp-20-11349-2020, 2020.

Zhou, S. S., Tai, A. P. K., Sun, S., Sadiq, M., Heald, C. L., and Geddes, J. A.: Coupling between surface ozone and leaf area index in a chemical transport model: strength of feedback and implications for ozone air quality and vegetation health, Atmos. Chem. Phys., 18, 14133–14148, https://doi.org/10.5194/acp-18-14133-2018, 2018.

Zhu, J., Tai, A. P. K., and Yim, S. H. L.: Effects of ozone-vegetation interactions on meteorology and air quality in China using a two-way coupled land-atmosphere model, Atmos. Chem. Phys. Discuss. [preprint], https://doi.org/10.5194/acp-2021-165, in review, 2021.

**References**

- Curci, G., Beekmann, M., Vautard, R., et al. Modelling study of the impact of isoprene and terpene biogenic emissions on European ozone levels. Atmospheric Environment[J]: 2009, 43: 1444-1455.
- Farquhar, G. D., von Caemmerer, S. v. and Berry, J. A. A biochemical model of photosynthetic CO2 assimilation in leaves of C3 species. Planta[J]: 1980, 149: 78-90.
- Feng, Z. Z., Yuan, X. Y., Fares, S., et al. Isoprene is more affected by climate drivers than monoterpenes: A meta-analytic review on plant isoprenoid emissions. Plant Cell and Environment[J]: 2019, 42: 1939-1949.
- Fitzky, A. C., Sandén, H., Karl, T., et al. The interplay between ozone and urban vegetation–BVOC emissions, ozone deposition and tree ecophysiology. Frontiers in Forests and Global Change[J]: 2019, 2: 50.
- Gong, C., Lei, Y., Ma, Y., et al. Ozone-vegetation feedback through dry deposition and isoprene emissions in a global chemistry-carbon-climate model. Atmospheric Chemistry and Physics[J]: 2020, 20: 3841-3857.
- Heald, C. L. and Geddes, J. A. The impact of historical land use change from 1850 to 2000 on secondary particulate matter and ozone. Atmospheric Chemistry and Physics[J]: 2016, 16: 14997-15010.
- Huntingford, C., Oliver, R. J., Mercado, L. M., et al. Technical note: A simple theoretical model framework to describe plant stomatal "sluggishness" in response to elevated ozone concentrations. Biogeosciences[J]: 2018, 15: 5415-5422.
- Lombardozzi, D., Levis, S., Bonan, G., et al. Predicting photosynthesis and transpiration responses to ozone: decoupling modeled photosynthesis and stomatal conductance. Biogeosciences[J]: 2012, 9: 3113-3130.
- Pacifico, F., Folberth, G., Sitch, S., et al. Biomass burning related ozone damage on vegetation over the Amazon forest: a model sensitivity study. [J]: 2015, 2015.
- Paoletti, E., Grulke, N. E. and Matyssek, R. Ozone Amplifies Water Loss from Mature Trees in the Short Term But Decreases It in the Long Term. Forests[J]: 2019, 11: 46.
- Pope, R. J., Arnold, S. R., Chipperfield, M. P., et al. Substantial increases in Eastern Amazon and Cerrado biomass burning-sourced tropospheric ozone. Geophysical Research Letters[J]: 2020, 47: e2019GL084143.
- Sadiq, M., Tai, A. P. K., Lombardozzi, D., et al. Effects of ozone-vegetation coupling on surface ozone air quality via biogeochemical and meteorological feedbacks. Atmospheric Chemistry and Physics[J]: 2017, 17: 3055-3066.
- Sitch, S., Cox, P. M., Collins, W. J., et al. Indirect radiative forcing of climate change through ozone effects on the land-carbon sink. Nature[J]: 2007, 448: 791-794.
- Spitters, C. Separating the diffuse and direct component of global radiation and its implications for modeling canopy photosynthesis Part II. Calculation of canopy photosynthesis. Agricultural and Forest meteorology[J]: 1986, 38: 231-242.
- Von Caemmerer, S. v. and Farquhar, G. D. Some relationships between the biochemistry of photosynthesis and the gas exchange of leaves. Planta[J]: 1981, 153: 376-387.
- Wittig, V. E., Ainsworth, E. A., Naidu, S. L., et al. Quantifying the impact of current and future tropospheric ozone on tree biomass, growth, physiology and biochemistry: a quantitative meta-analysis. Global Change Biology[J]: 2009, 15: 396-424.

- Yue, X. and Unger, N. The Yale Interactive terrestrial Biosphere model version 1.0: description, evaluation and implementation into NASA GISS ModelE2. Geoscientific Model Development[J]: 2015, 8: 2399-2417.
- Yue, X. and Unger, N. Fire air pollution reduces global terrestrial productivity. Nat Commun[J]: 2018, 9: 5413.
- Zhou, S. S., Tai, A. P. K., Sun, S. H., et al. Coupling between surface ozone and leaf area index in a chemical transport model: strength of feedback and implications for ozone air quality and vegetation health. Atmospheric Chemistry and Physics[J]: 2018, 18: 14133-14148.
- Zhu, J., Tai, A. P. K. and Yim, S. H. L. Effects of ozone-vegetation interactions on meteorology and air quality in China using a two-way coupled land-atmosphere model. Atmos. Chem. Phys. Discuss. [preprint][J]: 2021, doi: https://doi.org/10.5194/acp-2021-165, 2021.